# Inflammasomes as Targets for Adjuvants

**DOI:** 10.3390/pathogens9040252

**Published:** 2020-03-30

**Authors:** Konstantin Ivanov, Ekaterina Garanina, Albert Rizvanov, Svetlana Khaiboullina

**Affiliations:** 1Kazan Federal University, 420008 Kazan, Russia; KYIvanov@kpfu.ru (K.I.); EEGaranina@kpfu.ru (E.G.); Albert.Rizvanov@kpfu.ru (A.R.); 2Shemyakin-Ovchinnikov Institute of Bioorganic Chemistry, Russian Academy of Sciences, 117997 Moscow, Russia; 3University of Nevada, Reno, NV 89557, USA

**Keywords:** vaccines, adjuvants, inflammasomes, innate immunity, cytokines

## Abstract

Inflammasomes are an essential part of the innate immune system. They are necessary for the development of a healthy immune response against infectious diseases. Inflammasome activation leads to the secretion of pro-inflammatory cytokines such as IL-1β and IL-18, which stimulate the adaptive immune system. Inflammasomes activators can be used as adjuvants to provide and maintain the strength of the immune response. This review is focused on the mechanisms of action and the effects of adjuvants on inflammasomes. The therapeutic and prophylaxis significance of inflammasomes in infectious diseases is also discussed.

## 1. Introduction

Vaccination is the most effective way to prevent infectious diseases [1]. There are several different types of vaccines available, including live attenuated and inactivated vaccines, subunit and toxoid vaccines, virus-like particle (VLP) based vaccines, as well as polysaccharide and polysaccharide conjugate vaccines [2]. Vaccines containing live or whole killed pathogens can induce strong protective immune responses without adjuvants. However, on rare occasions these vaccines can revert to a virulent strain through back-mutation, compensatory mutations, recombination or reassortment [3]. This can cause disease and side effects as a result of immunization, which is not characteristic of other vaccines [4]. In contrast, subunit, recombinant, polysaccharide, and conjugated vaccines are generally safe, but they are less effective without adjuvants. Adjuvants can be an organic or non-organic derivative [5]. Some of them are naturally occurring, for instance, mineral salts of aluminum or bacterial components that are used in complete Freund’s adjuvant (CFA) [6]. In contrast, subunit vaccines have limited immunogenicity, mostly because they lack natural adjuvants [7]. Therefore, currently, almost every vaccine contains adjuvants, which enhance and prolong the immune response [8]. 

There are several adjuvants that have been approved for human application, including aluminum salts, emulsions such as MF-59, and AS03 [9]. For most of these adjuvants, the mechanism of action remains unclear [10]. Among adjuvants, aluminum salts are the most studied. It is believed that aluminum salts might act as a matrix for antigen adsorption [11]. Adsorbed antigens appear to be more stable, which is crucial for them to retain immunogenicity [12]. A similar mechanism of action was suggested for incomplete Freund’s adjuvant (IFA), which can establish a depot at the site of injection for slow antigen release [13]. Some adjuvants also act through activation of toll-like receptors‘ (TLRs) [14]. For example, CFA, monophosphoryl Lipid A (MPL), flagellin and adjuvants based on CpG are ligands for TLR 2, 4, 5 and 9, respectively [15,16,17,18]. TLR agonists were shown to be effective adjuvants [19]. CFA appears to have the highest adjuvant activity compared to other adjuvants [20,21]. However, CFA has been shown to be associated with numerous side effects such as granulomas, ulcerative necrosis and sterile abscesses at the injection site [22].

Most of these adjuvants are still undergoing clinical trials, while aluminum salts remain a gold standard [23]. An ideal adjuvant should meet specific requirements such as safety, bio-degradability, and long-term stability. Most of the approved adjuvants do not fully comply with these criteria [9]. Therefore, the search for and development of new adjuvants is urgently needed to improve vaccines. The discovery of inflammasome structure and function initiated a new age in the development of vaccines and adjuvants. It has been demonstrated that inflammasomes, which can recognize damage-associated molecular patterns (DAMPs), are also involved in the mechanisms of adjuvant action [24].

## 2. Innate Immunity

The immune response can be divided into innate and adaptive [25]. For innate immunity, all receptors expressed by cells are encoded in the genome and passed to the next generation, unlike receptors of the adaptive immune system, which are formed as a result of V(D)J rearrangement and somatic hypermutation [26]. The innate immune system is characterized by its capacity to recognize a wide range of pathogens including viruses, fungi, and bacteria through the number of pattern recognition receptors (PRRs) [18]. PRRs recognize conservative microbial signals, the so-called pathogen-associated molecular patterns (PAMPs) [27]. The activated innate immune system elicits an inflammatory response followed by production of cytokines and chemokines to attract immune cells to the place of infection and create an adaptive immune response [28]. PRRs include TLRs, RIG-1-like receptors (RLRs) and pattern recognizing NOD-like receptors (NLRs) [27]. NLRs are the key sensors in the innate immune response and are one of the main inflammasome components involved in sensing pathogens [29].

## 3. Inflammasomes

Inflammasomes represent a group of cytosolic multiprotein complexes composed of a sensor protein belonging to the NLR or Pyrin and HIN domain families (PYHIN), a caspase-1 effector and an adaptor protein ASC [30]. The structure of inflammasome is shown in Figure 1.

Inflammasomes are expressed in various cells, including granulocytes, Т- and В-cells, monocytes, hepatocytes, neurons, microglia, and Langerhans cells [31,32,33]. The functions of inflammasomes include sensing the pathogen and launching a subsequent innate immune response [34,35]. Inflammasomes are formed when PAMP and DAMP are sensed in the cytoplasm [36]. This induces conformational changes in the pre-assembled NLRP3-inactive inflammasome [37]. Finally, activated inflammasome proteolytically cleaves pro-caspase 1, releasing the active caspase 1 [38]. Next, the caspase-1 digests pro-interleukin-1-beta (IL-1β) and pro-interleukin-18 (IL-18) to their active forms, which are secreted by a cell [39,40]. These interleukins are involved in establishing the inflammation and immune response that protect against microorganisms.

IL-18 can activate lymphocytes [41,42]. Also, it can increase Т- and B-cell proliferation, enhance the activity of natural killers (NKs), stimulate secretion of interferon-gamma (IFNγ), tumor necrosis factor-alpha (TNFα), interleukin-1 (IL-1) and interleukin-2 (IL-2) [43,44]. IL-18 induced release of TNFα and IL-1 appears to depend on inflammasome activation [45]. Another inflammasome product, IL-1β, seems to stimulate the proliferation of Т-cells [46,47,48]. Also, IL-1β may increase the number of CD103+CD69+ tissue-resident memory Т-cells (TRM) and dendritic cells [49]. It appears that IL-18 and IL-1β, which are products of activated inflammasomes, have a similar effect on leukocytes, that is, they stimulate proliferation and promote immune competence [50,51,52]. Moreover, some cytokines can be administered as adjuvants. IL-6, IL-1 and IL-12 promote either Th2- or Th1-type responses, thereby enhancing systemic immunity to co-administered antigens. IL-18 enhances the effects of IL-12 in inducing an antigen-specific Th1 type CD4+ T cell response as well as high titers of IgG antibodies [53,54].

## 4. Adjuvants as Inflammasome Activators

Most adjuvants have similar mechanisms of inflammasome activation, including lysosome degradation and cathepsine release and formation of reactive oxygen species, which subsequently lead to IL-1β and IL-18 secretion (Figure 2 and Figure 3).

### 4.1. Aluminum Adjuvants

Aluminum-containing adjuvants were first used in human vaccines in 1932 [55]. Then, they were applied in numerous studies, where their efficiency as adjuvants was confirmed. Currently, aluminum adjuvants such as aluminium potassium sulphate, aluminium hydroxide, aluminium phosphate, and amorphous aluminium hydroxyphosphate sulfate are considered to be the gold standard for newly developing adjuvants [56]. Interestingly, the most commonly used adjuvants, such as aluminum salts, are capable of activation of NLRP3 inflammasome [57]. It has been shown that aluminum hydroxide-based adjuvants can stimulate secretion of pro-inflammatory cytokines IL-1β and IL-18 by activating NLRP3 inflammasome [58,59]. A mechanism of inflammasome activation was explained by the lysosome damage induced by aluminum hydroxide, which triggers the release of cathepsin. Free cathepsin B can initiate the inflammasome assembly, which activates caspase-1 and releases active cytokines [60,61]. Initial activation of a nuclear transcription factor (NF-κB) via lipopolysaccharide (LPS) appears to be required for NLRP3 activation by aluminum hydroxide [62]. Also, aluminum hydroxide can increase the uric acid levels at the site of vaccination, which could promote pro-IL-1b and pro-IL-18 conversion into their active forms by activated NLRP3 inflammasome [63]. 

### 4.2. Chitosan

Natural substances have also been shown to possess adjuvant activity. For example, chitosan, a highly biodegradable chitin derivative can enhance both humoral and cell-mediated immune response, which is equipotent to IFA and superior to aluminum hydroxide [64]. Chitosan can enter the cell by phagocytosis and activate NLRP3 inflammasome [65]. Beuter et al. have demonstrated that the mechanism of inflammasome activation is associated with potassium ion efflux and lysosome destabilization, as well as the formation of reactive oxygen species (ROS) [66]. Also, chitosan, used as an influenza vaccine adjuvant, was shown to increase the production of IL-2, IL-4, IL-6, TNF, IL-17A, and IL-10. Increased IFNγ production and IgG titers indicate the enhancement of the cell-mediated and humoral immune responses by chitosan adjuvant [67].

### 4.3. Saponins

Saponins isolated from the soapwort plant are also known to enhance an immune response [68]. Kensil et al. were the first to isolate saponin fractions from the soapwort plant with various degrees of adjuvant properties [69]. Adjuvants based on the QS-21 portion of saponin had low toxicity combined with excellent adjuvant effects. They were used in more than 100 human clinical trials [70,71] where they demonstrated the ability to induce humoral and cell-mediated immune response against a wide range of antigens [72]. Also, the QS-21 fraction was superior to aluminum salts because they induced a Th1-type immune response, which is essential for controlling intracellular pathogens [73]. 

The mechanism of the QS-21 adjuvant activity is being investigated. Recent studies have confirmed that QS-21 activates NLPR3 inflammasome [74]. An arrangement of NLPR3 inflammasome activation by QS-21 was linked to lysosome destabilization and cathepsin B release [75]. Interestingly, QS-21 can correctly activate NLRP3 inflammasome in CD169+ macrophages localized in lymph nodes [76] and increase serum levels of IL-2 and IFNγ [77].

### 4.4. Synthetic Cation Polymeric Adjuvants

Progress in organic synthesis methods has contributed to the generation of artificial molecules with adjuvant activity. One of these is synthetic cation polymeric adjuvants containing biodegradable particles, which form nanoparticles [78]. Nanoparticles based on poly-d,l-lactide-co-glycolide, cationic lipids possess adjuvant properties that could be mediated by activation of the NLRP3 inflammasome [79]. The mechanism of inflammasome activation appears to be related to lysosome disruption and cathepsin B release as well as ROS formation [80]. 

These nanoparticle adjuvants were tested in animal models where they increased IgG titer and the level of TNF-α, INFγ, IL-17A, and IL-1β cytokines [81,82]. 

### 4.5. Cholera Toxin B

Cell wall components, some endotoxins, as well as nucleic acids are also capable of inflammasome activation. Therefore, they might have potential utility in adjuvant applications [83]. Cholera toxin, a protein of *Vibrio cholerae*, consists of six subunits, including one toxin A and five toxin B [84]. Cholera toxin B is a non-toxic protein with proven adjuvant efficacy, especially for mucosal vaccines [85,86]. It can also bind to GM1 ganglioside receptor and enter the cell, thereby activating NLRP3 and other inflammasomes [87]. The mechanism of activation appears to be related to the cholera toxin B enhancement of the small Rho GTPases (RhoA) activity via protein kinase A [88]. As a result, a pyrin receptor interacts with modified RhoA and triggers inflammasome self-assembly [89]. The efficacy of cholera toxin B as an adjuvant was demonstrated in a mouse model [90]. This adjuvant was able to increase the circulating IgG titers as well as mucosal IgA levels. Cholera toxin B also enhanced Т-cell proliferation and increased IL-17A and IFNγ production [91]. 

### 4.6. Flagellin

Flagellin, which forms the hollow filaments in bacterial flagellum, represents bacterial pathogenicity and virulence factor [92]. Several studies have demonstrated flagellin adjuvant activity in the mucosal vaccination [93,94,95]. Like LPS, it is a natural TLR agonist and can also trigger inflammasome assembly [96,97,98]. However, in contrast to LPS, flagellin induces formation of both NLRP3 and NLRC4 inflammasomes [99,100]. The efficacy of the flagellin as an adjuvant was demonstrated in experiments with mice and primates in a flu model [101,102]. Flagellin was shown to induce the production of various pro-inflammatory cytokines such as IL-6, IL-8, and CXCL2, which can facilitate a productive immune response [103,104,105]. It has been demonstrated that flagellin activates inflammasomes through TLR5 or NLRC4 receptors [106]. According to Dos Reis et al., the NLRP3 receptor function is not absolutely required for the flagellin adjuvant activity. For example, it was demonstrated that flagellin can stimulate the assembly of another inflammasome, NLRC4 in cells with defective NLRP3 receptors [100]. While flagellin activates NLRC4 inflammasome directly via the NAIP5-receptor, the formation of NLRP3 inflammasome appears to be indirectly regulated via cathepsin B by destabilizing lysosomes [107,108].

### 4.7. Nucleic Acids

Nucleic acids are traditionally accepted as a genetic vaccine; however, they can potentially be used as adjuvants. Viral single and double-stranded RNAs are recognized by the intracellular receptor RIG-1 (retinoic acid-inducible gene I), which can subsequently activate NLRP3 inflammasome [109]. Therefore, RNA appears to be useful as both a genetic vaccine epitope and an adjuvant [110]. These adjuvants have been shown to cause low toxicity and adverse effects [111]. Also, non-translational RNA can enhance an immune response [112]. For example, Doener et al. demonstrated that both cell-based and humoral immune responses were significantly increased in healthy volunteers who received a licensed rabies vaccine containing non-translational RNA adjuvant, CV8102 [113]. 

Double-stranded DNA could be detected by the AIM2 inflammasome [114], triggering inflammasome assembly and increasing IL-1β and IL-18 secretion [115,116]. Therefore, plasmids which are utilized as genetic vaccines can activate AIM2 inflammasome and enhance cell-based and humoral immune responses, even if they do not code pathogenic proteins [117,118].

## 5. Conclusions

Inflammation is an important component of the immune response, as it plays a role in stimulating leukocyte proliferation and differentiation, releases regulatory cytokines and activates non-specific immunity [119]. Inflammation could be regulated by many pathways, including inflammasomes [120]. As inflammasome senses PAMPs and DAMPs, it starts forming multiprotein complexes, and releasing active caspase 1, which, in turn, cleaves IL-1β and IL-18 [36]. Both cytokines play essential roles in establishing and maintaining inflammation [39,43,44,46,48,49,50,51,52]. Also, these cytokines directly stimulate immune cells like T- and B-lymphocytes and dendritic cells [41,42,47]. Inflammation-caused leukocyte activation and migration to the site of infection contributes to antigen recognition and presentation [121]. 

Therefore, activation of inflammasomes as part of immune recognition during vaccination could help facilitate antigen recognition and presentation. Inflammasomes could be targeted by adjuvants [122]. As adjuvants help to induce a strong and long-lasting immune response, targeting inflammasomes could be a potential mechanism of their action. Adjuvants are widely utilized in vaccines and their effect can directly depend on their capacity to activate the inflammasome. In this review we summarized information on adjuvants currently used for vaccination and their effect on inflammasome activation. Several adjuvants, aluminum salts, MF-59, and AS03 have been shown to activate inflammasome as part of their mechanism of action. It appears that inflammasome activation facilitates the development of a strong and long-lasting immune response [34,35]. This was shown for aluminum salts, MF-59, and AS03 adjuvants, which are currently used as part of influenza, polio, hepatitis A and B vaccines [123,124,125,126,127]. Using inflammasomes as a target for adjuvants mimics the innate immune response to an infectious agent and increases the effectiveness of the vaccine. Therefore, targeting molecular mechanisms involved in inflammasome activation could be beneficial for the development of new approaches for adjuvant‘ design and to further advance the development of effective vaccine formulations.

## Figures and Tables

**Figure 1 pathogens-09-00252-f001:**
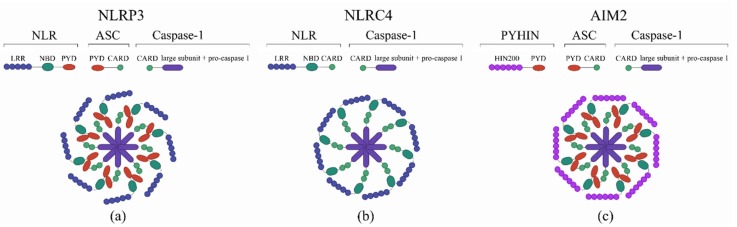
The structure of the inflammasome. (**a**) NOD-like receptor protein 3 (NLRP3) inflammasome, (**b**) NLR-family CARD domain-containing protein 4 (NLRC4) inflammasome, (**c**) Absent in melanoma 2 (AIM2) inflammasome. The inflammasome structure is composed of functional units such as a leucine-rich repeat (LRR) C-terminal or DNA-binding domain (HIN200), a nucleotide binding domain (NBD), a pyrin domain (PYD), and a caspase recruitment domain (CARD).

**Figure 2 pathogens-09-00252-f002:**
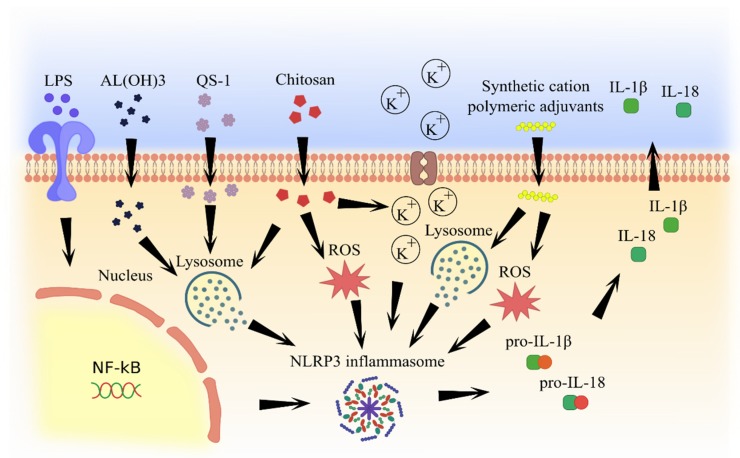
Molecular pathways of inflammasome activation by natural and artificial (synthetic) adjuvants.

**Figure 3 pathogens-09-00252-f003:**
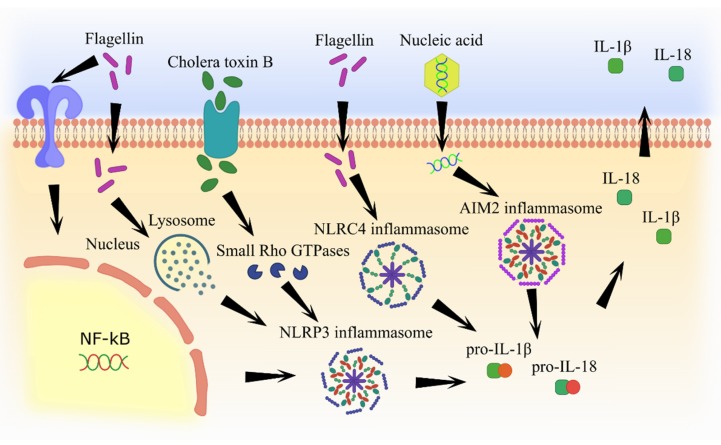
Molecular pathways of inflammasome activation by pathogen component-based adjuvants.

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
