# Peer review of "Inflammasomes as Targets for Adjuvants"

_pathogens, 2020, doi:10.3390/pathogens9040252_

Round 1
Reviewer 1 Report
General: The authors have a tendency to put “the” in sentences where it is not needed or wanted. example: Line 44, 46, 53, 54, 84
Line 2: I would suggest a change to the title: Inflammasomes as targets for adjuvants
line 13: Recommend rewording sentence “Nowadays, activators of inflammasomes are being increasingly used in medicine like adjuvants, which are auxiliary components of vaccines, providing the continued strength of the immune response”
Line 23: change particles to particle
Line 24-25: the authors talk about live attenuated and whole cell vaccines and then say they are associated with adjuvants? These vaccines are not adjuvanted but rather the components are highly reactogenic leading to the very strong immune response. These sentences should be re-worded or re-structured as I’m sure they are referring to adjuvanted subunit vaccines in the later statement but it is very unclear as submitted.
Line 32: Application to Applications
Line 33: Freund's adjuvant and CFA are not approved for human use and you are missing CPG. I would reference: Petrovsky N. Comparative Safety of Vaccine Adjuvants: A Summary of Current Evidence and Future Needs. Drug Saf. 2015;38(11):1059–1074. doi:10.1007/s40264-015-0350-4
And
McIntire CR, Yeretssian G, Saleh M. Inflammasomes in infection and inflammation. Apoptosis. 2009;14(4):522–535. doi:10.1007/s10495-009-0312-3.
Possibly also discuss Flagellin? (Dos Reis EC, et al. Front Immunol 2019. PMID 31244842)
I can not find reference 8. Perhaps a better reference here would be Del Giudice G, Rappuoli R, Didierlaurent AM. Correlates of adjuvanticity: A review on adjuvants in licensed vaccines. Semin Immunol. 2018;39:14–21. doi:10.1016/j.smim.2018.05.001 (your reference 15)
Line 40: please explain the side effects
Line 50: Paragraph does not read well including the innate immunity and the contact (contact of what?). please rephrase
Line 55 Line 56: either “receptors are” or “receptor is” and “part of the innate”
Line 60: define PYHIN
Line 67: change to Inflammasome function
Line 69: Define PAMP
Line 71: Finally,
Line 84: Is this true? CPG, TLR4 do not activate the inflammasome.
Line 87: Aluminum phoshate is also widely used. Perhaps it should be mentioned?
Author Response
Please see the attachment
General: The authors have a tendency to put “the” in sentences where it is not needed or wanted. example: Line 44, 46, 53, 54, 84
Agree: changes were made in the manuscript
Line 2: I would suggest a change to the title: Inflammasomes as targets for adjuvants
Agree: the title was changed to: Inflammasomes as targets for adjuvants
line 13: Recommend rewording sentence “Nowadays, activators of inflammasomes are being increasingly used in medicine like adjuvants, which are auxiliary components of vaccines, providing the continued strength of the immune response”
Agree: sentence was changed to: “Inflammasomes activators can be used as adjuvants, thus providing the continued strength of the immune response.”
Line 23: change particles to particle
Agree (new line 22): changes were made: “particles” was changed to “particle”
Line 24-25: the authors talk about live attenuated and whole cell vaccines and then say they are associated with adjuvants? These vaccines are not adjuvanted but rather the components are highly reactogenic leading to the very strong immune response. These sentences should be re-worded or re-structured as I’m sure they are referring to adjuvanted subunit vaccines in the later statement but it is very unclear as submitted.
Agree (new lines 23-28): sentence was changed to “Vaccines containing live or whole killed pathogens can induce strong protective immune responses without adjuvants. However, in rare occasion these vaccines could reverse to a virulent strain through back-mutation, compensatory mutations, recombination or reassortment [3]. This could cause the disease and side effects as the result of immunization, which is uncharacteristic to other vaccines [4]. In contrast, subunit, recombinant, polysaccharide, and conjugated vaccines are generally safe, but they are less effective without adjuvants.”
Line 32: Application to Applications
Agree (new line 34): changes were made in line 32: “application” to “applications”
Line 33: Freund's adjuvant and CFA are not approved for human use and you are missing CPG. I would reference: Petrovsky N. Comparative Safety of Vaccine Adjuvants: A Summary of Current Evidence and Future Needs. Drug Saf. 2015;38(11):1059–1074. doi:10.1007/s40264-015-0350-4
And
McIntire CR, Yeretssian G, Saleh M. Inflammasomes in infection and inflammation. Apoptosis. 2009;14(4):522–535. doi:10.1007/s10495-009-0312-3.
Agree (new line 34): The statement that “ Freund's adjuvant and CFA are approved for human use” was removed.
Possibly also discuss Flagellin? (Dos Reis EC, et al. Front Immunol 2019. PMID 31244842)
Agree (new line 41): Sentences were added: “Some adjuvants also act through activation of toll-like receptors` (TLRs) [14]. For example, CFA, monophosphoryl Lipid A (MPL), flagellin and adjuvants based on CpG are ligands for TLR 2, 4, 5 and 9, respectively [15–18]. TLR agonists were shown to be effective adjuvants [19].”
I can not find reference 8. Perhaps a better reference here would be Del Giudice G, Rappuoli R, Didierlaurent AM. Correlates of adjuvanticity: A review on adjuvants in licensed vaccines. Semin Immunol. 2018;39:14–21. doi:10.1016/j.smim.2018.05.001 (your reference 15)
Agree (new line 35): reference 8 was replaced by Del Giudice G, Rappuoli R, Didierlaurent AM. Correlates of adjuvanticity: A review on adjuvants in licensed vaccines. Semin Immunol. 2018;39:14–21.
Line 40: please explain the side effects
Agree (new lines 43-45): sentence was clarified: “However, CFA was shown to be associated with numerous side effects such as granulomas, ulcerative necrosis and sterile abscesses in the injection site [22].”
Line 50: Paragraph does not read well including the innate immunity and the contact (contact of what?). please rephrase
Agree (new lines 58-65): sentence was changed to: “Innate immune system is characterized with capacity for recognition wide range of pathogens including viruses, fungi, bacteria, through the restricted amount of pattern recognition receptors (PRRs) [18]. PRRs recognize conservative microbial signals so called pathogen-associated molecular patterns (PAMPs) [27]. While activated innate immune system elicits inflammatory response producing cytokines and chemokines for attraction of immune cells to the place of infection and cause adaptive immune response [28]. PRRs include TLRs, RIG-1 – like receptors (RLRs) and pattern recognizing NOD-like receptors (NLRs) [27]. NLRs are the key part of the innate immune recognition and one of the main inflammasome components sensing the pathogen [29].”
Line 55 Line 56: either “receptors are” or “receptor is” and “part of the innate”
Agree (new line 64-65): Changes were made in line 55 and 56 to “NLRs are the key part of the innate immune recognition and one of the main inflammasome components sensing the pathogen [29].”
Line 60: define PYHIN
Agree (new line 70): abbreviation of PYHIN was defined as: “Pyrin and HIN domain families”
Line 67: change to Inflammasome function
Agree (new line 77): changes were made: to “Inflammasome function”
Line 69: Define PAMP
Agree (new line 61): definition of PAMP was provided as “pathogen-associated molecular patterns”
Line 71: Finally,
Agree (new line 81): changes were made in line 71 to “Finally,”
Line 84: Is this true? CPG, TLR4 do not activate the inflammasome.
Agree that they are not activate inflammasomes but they are required for inflammasome priming. Inflammasome priming is the first step for inflammasome activation. Therefore we included CpG and TLR4 agonists in this review.
We made following changes in the text according to your comments (new line 40-45) “Some adjuvants also act through activation of toll-like receptors` (TLRs) [14]. For example, CFA, monophosphoryl Lipid A (MPL), flagellin and adjuvants based on CpG are ligands for TLR 2, 4, 5 and 9, respectively [15–18]. TLR agonists were shown to be effective adjuvants [19]. CFA appears to have the highest adjuvant activity as compare to other adjuvants [20,21]. However, CFA was shown to be associated with numerous side effects such as granulomas, ulcerative necrosis and sterile abscesses in the injection site [22].”
Line 87: Aluminum phoshate is also widely used. Perhaps it should be mentioned?
Agree (new lines 104-105): changes were made: the title of the section was changed to “Aluminum adjuvants”. Also, sentence: “Currently, aluminum adjuvants are considered as a golden standard for newly developing adjuvants.” was changed to: “Currently, aluminum adjuvants such as aluminium potassium sulphate, aluminium hydroxide, aluminium phosphate, and amorphous aluminium hydroxyphosphate sulfate are considered as a golden standard for newly developing adjuvants [56].”
Reviewer 2 Report
In this review manuscript by Ivanov and colleagues, the authors focus on the inflammasome as the main target of currently and under development adjuvants and discuss their importance in eliciting an effective immune response following vaccination.
The review article is of importance for the scientific community working in the field. However, the authors would need to address the following criticisms before considering the manuscript for publication in Pathogens.
- There are some typos and spell errors in the whole manuscript. I would recommend the authors to carefully review and amend it accordingly. As an example:
- Line 70 should amend as “conformational”
- Line 118 amend as “a wide range”
Additionally, the authors can also discuss the co-administration of cytokines along with vaccines as a possible adjuvant-based strategy.
Furthermore, the author should expand their conclusions. In its current form, the manuscript lacks substantial conclusions.
Author Response
Please see the attachment
In this review manuscript by Ivanov and colleagues, the authors focus on the inflammasome as the main target of currently and under development adjuvants and discuss their importance in eliciting an effective immune response following vaccination.
The review article is of importance for the scientific community working in the field. However, the authors would need to address the following criticisms before considering the manuscript for publication in Pathogens.
- There are some typos and spell errors in the whole manuscript. I would recommend the authors to carefully review and amend it accordingly. As an example:
Line 70 should amend as “conformational”
Agree (new line 80): change was made to “conformational”
Line 118 amend as “a wide range”
Agree (new line 133): changes were made to “a wide range”
Additionally, the authors can also discuss the co-administration of cytokines along with vaccines as a possible adjuvant-based strategy.
Agree (new lines 92-96): discussion of the effect of cytokine and vaccine co-administration was added in section “Moreover, some cytokines can be administered as adjuvants. IL-6, IL-1 and IL-12 promote either Th2- or Th1-type responses, thereby enhance systemic immunity to co-administered antigens. IL-18 enhances the effects of IL-12 in inducing an antigen-specific Th1 type CD4+ T cell response as well as high titers of IgG antibodies [53,54].”
Furthermore, the author should expand their conclusions. In its current form, the manuscript lacks substantial conclusions.
Agree (new lines 194-216): changes were made in conclusion: “Inflammation is an important component of the immune response, playing role in stimulating leukocyte proliferation and differentiation, releasing regulatory cytokines and activating non-specific immunity [119]. Inflammation could be regulated by many pathways, including inflammasomes [120]. As inflammasome senses PAMPs and DAMPs, it starts forming multiprotein complex, releasing active caspase 1, which, in turn, cleaves IL-1β and IL-18 [36]. Both cytokines play essential role in establishing and maintaining inflammation [39,43,44,46,48–52]. Also, these cytokines directly stimulate immune cells like T and B lymphocytes and dendritic cells [41,42,47]. Inflammation caused leukocyte activation and migration to the site of infection contributes to the antigen recognition and presentation [121].
Therefore, activation inflammasomes as part of the immune recognition during vaccination could help facilitate antigen recognition and presentation. Inflammasomes could be targeted by adjuvants [122]. As adjuvants assist to induce strong and long lasting immune response, targeting inflammasomes could be a potential mechanism of their action. Existing adjuvants are widely utilized in vaccine and their effect directly depends on capacity activate the inflammasome. In this review we summarized information on adjuvants currently used for vaccination and their effect on inflammasome. Several adjuvants, aluminum salts, MF-59, and AS03 have shown to engage NALP3 inflammasome activation as part of their action. It appears that inflammasome activation facilitates development of strong and long lasting immune response [34,35]. This was shown for aluminum salts, MF-59, and AS03 adjuvants, which are currently used as part of influenza, polio, hepatitis A and B vaccines [123–127]. Using inflammasomes as a target for adjuvants mimics the innate immune response to an infectious agent and increases the effectiveness of the vaccine. Therefore, targeting molecular mechanisms involved in inflammasome activation could be beneficial for development of new approaches for adjuvants` design and further efficient vaccine development.”
Round 2
Reviewer 1 Report
I hope the editor will supply you with the word version of my suggested edits in tracked changes. I feel most of my major concerns from the first review have been addressed but many of the grammatical errors have remained.

Author Response
We have revised all the grammatical errors. Please see the attachment
